# Vertical Integration in the Pediatrics Clerkship: A Case Study

**Julianne M. Hall** [1,*], **Rahul Anand** [1], **Lynn E. Copes** [1], **Kelly Moran-Crusio** [1], **J. Nathan Davis** [1], **Maya Doyle** [2], **Christine Maffeo** [1] and **Eitan S. Kilchevsky** [1]

[1] Frank H. Netter MD School of Medicine, Quinnipiac University, Hamden, CT 06518, USA;
rahul.anand@quinnipiac.edu (R.A.); lynn.copes@quinnipiac.edu (L.E.C.);
kelly.crusio@quinnipiac.edu (K.M.-C.); nathan.davis@quinnipiac.edu (J.N.D.);
christine.maffeo-anton@quinnipiac.edu (C.M.); eitan.kilchevsky@quinnipiac.edu (E.S.K.)

[2] Department of Social Work, School of Health Sciences, Quinnipiac University, Hamden, CT 06518, USA;
maya.doyle@quinnipiac.edu

* Correspondence: julianne.hall@quinnipiac.edu

**Abstract:** Since the end of the twentieth century, medical educators continue to review and call for changes that will improve how medical students apply their knowledge of basic sciences to the clinical management of their patients. The traditional 2 + 2 curriculum, where basic sciences are taught during the first two years and were followed by clinical clerkships, was challenged with calls to move towards a Z-shaped integrated curriculum, a model which presents bio-medical sciences and clinical cases in parallel or in connection with one another. Faculty at the Frank H. Netter MD School of Medicine developed a vertical integration didactic session that presented an eight-year-old child with an acute asthmatic episode. After a brief introduction, clinical and pre-clinical faculty who teach in Years 1–3 and social work faculty met with medical students placed in small groups to discuss their pertinent field; faculty members rotated among the groups. At the end of the session, the students provided feedback and comments for the continuous quality improvement of the session. The session has been taught four times thus far. A majority of the students expressed satisfaction with the opportunity to review basic science concepts during the clerkship and apply these concepts to develop clinical management skills. Students were also excited to discuss social determinants and the effects of a pediatric chronic illness on the whole family. Combining a review of basic and social science concepts with clinical management, with faculty from pre-clinical and clinical years, was enjoyed by our students, who felt this educational approach expanded their ability to better manage clinical problems. While our case is in pediatrics, we believe the method can be applied to other specialties.

**Keywords:** vertical integration; integrated curriculum; Z-shaped curriculum; learning objectives; asthma; social determinants of health





## 1. Introduction

The traditional medical school curriculum based on 2 + 2 years requires memorizing significant amount of basic sciences information and a large volume of clinical work. This traditional curriculum went through many reviews since the Flexner Report [1,2]. It seems, though, that since the introduction of the term "integrated curriculum", curricular changes in many schools have risen exponentially, as manifested by the number of articles discussing "integrated curriculum", rising from less than 5 in 1983 to 80 in 2013 [3].

Integration in education was defined by Harden as "the organization of teaching matter to interrelate or unify subjects frequently taught in separate academic courses or departments" [4]. In relevance to medical education, Dahle et al. suggested that vertical integration (VI) is "the integration between the clinical and basic science parts of the curriculum" [5]. Wijnen-Meijer further defined VI as a "deliberate educational approach that fosters a gradual increase of learner participation in the professional community

through a stepwise increase of knowledge-based engagement in practice with graduated responsibilities in patient care" and generating a genuine team member sensation [6]. Regardless of the definition, vertical integration (unlike horizontal integration) means integration across time—the time spent on classroom education gradually decreases across the four years of medical school, while the time spent in clinical practice increases, along with a blending of the solution of clinical problems with continuously teaching of basic science. Thus, the traditional 2 + 2 curriculum is being replaced with an innovative Z-shaped curriculum, where the traditional basic sciences are taught simultaneously with clinical cases [7] (Figure 1).

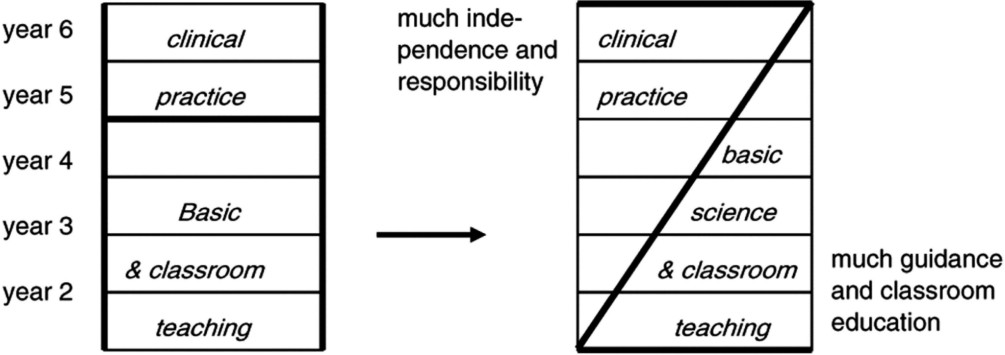

**Figure 1.** The traditional H-shaped medical curriculum replaced by Z-shaped medical curriculum [8]. On the left, the standard curriculum where the teaching of basic sciences precedes clinical education. On the right, the Z-shaped medical curriculum demonstrating progressive introduction to clinical practice while maintaining persistent basic and classroom teaching components throughout all years in medical school.

Data in the medical and educational literature suggests that an integrated curriculum helps establish a linkage between the subjects, addressing various areas so that the learning experience and retention are improved [9]. It was also suggested that a curriculum that only provides cognitive links with clinical practice, or opportunities to observe clinical practice, lacks the power of student engagement in patient care [10]. This led to the realization that there may be benefits to integrating clinical competencies while studying basic science. Wijnen-Meijer [7] indicated that vertical integration provides more meaningful early clinical experience, helps integrate biomedical sciences and clinical teaching, adds to the length of clerkships during the last years of medical school, and fosters responsibility within undergraduate training. Compared to graduates from a non-VI curriculum, the graduates from a VI curriculum make career choices earlier and need less time and fewer applications to obtain a residency position. The authors also suggested that these graduates feel better prepared for work and postgraduate training.

More support for an integrated curriculum was offered by Yardley et al., who studied the benefits of VI extensively. In a summary of the literature [11], they cited added benefits to the integrated curriculum, such as increase in students' motivation to learn, quicker development of clinical skills, help in considering career choices, learning about the role of doctors in different settings, better ability to communicate with patients, and understanding patient perspective. Knight and Mattick [12] added support to the concept of vertical integration in medical education by suggesting that developing the capacity and skill to integrate knowledge enabled students to progress from feeling overwhelmed by the need to retain numerous, discrete, discipline specific facts, to feeling confident of their ability to access that knowledge through the process of clinical reasoning. Given the wealth of research in the field, it was not surprising that the Liaison Committee on Medical Education (LCME), in its renewed standards, requires that a curriculum be "coherent and coordinated" and "integrated within and across the academic periods of study" [13].

With regards to trainees in Pediatrics, there have been major changes in the approach to medical education in the past century that are consistent with the need for a vertically

integrated curriculum. Changes began with the implementation of a substantial program of basic and social science with the recognition that these disciplines enable scientific reasoning that can be applied to clinical decision making [14]. Recommendations from a major Canadian medical symposium suggested that undergraduate medical education in pediatrics should also include the needs of vulnerable populations, early childhood development, preventative medicine, social determinants of health, mental/emotional/social health, and interdisciplinary care [15]. In the United States, the Accreditation Council for Graduate Medical Education further requires pediatric residents to demonstrate knowledge of established and evolving biomedical, clinical, epidemiological, and social-behavioral sciences, as well as the application of this knowledge to the patient [16]. As described in this case report, a vertical integration model in pediatrics enables trainees to integrate basic, clinical, and social-behavioral sciences, as well as interdisciplinary care, and to apply these domains to patient care. Specifically, the objective of this project was to enable 3rd year medical students in their pediatrics clerkship to vertically integrate basic and social science with clinical care in their approach to the longitudinal treatment of a pediatric patient.

## 2. Materials and Methods

In an attempt to help students integrate basic science knowledge with clinical management, members of the Quinnipiac University's Frank H. Netter MD School of Medicine (FHNSOM) faculty assembled and embarked on developing a new project: Vertical Integration in Pediatrics (VIP), an educational program. The group included FHNSOM professors of Anatomy (LC), Biochemistry (JND), Immunology (KC), Microbiology (RA), Pharmacology (JH), Physiology (JH), and Clinical Sciences (CM) and a faculty member from the Quinnipiac University School of Health Sciences/Department of Social Work (MD). The group also included the Director of the Pediatric Clerkship (EK) and the Pediatric Clerkship Coordinator (CL). The VIP program presents a pediatric clinical case, incorporating basic science elements while discussing the clinical course of the patient.

The case selected (Supplementary File S1) was that of an eight-year-old boy with asthma who is seen in the emergency department and, while being treated with bronchodilators and steroids, later develops hypotension. The case was structured to meet learning objectives in the following domains: Anatomy, Allergy/Immunology, Biochemistry, Microbiology, Pharmacology, Physiology, and Social and Family Determinants (see Figure 2).

Since the Vertical Integration in Pediatrics (VIP) event was novel for both the faculty and the students, the student evaluation form was developed to allow the further refinement of the learning event. To that end, the event was piloted with a small group of eight students who were currently participating in the pediatric clerkship.. The pilot group provided very favorable ratings for the learning experience, with all the students indicating that the event was worthwhile and seven of the eight students also providing positive comments. Additionally, the students unanimously agreed that the event increased their knowledge and that they welcome similar events in the future. The pilot group also solidly embraced the small group setting, with all eight students indicating that it was the preferred size (discussed further in Results). To gain further insight into the student experience, the students were asked to identify three valuable aspects. The narrative responses could be categorized into three major themes: appreciation for the integration between the pre-clinical and clinical content, valuable review of the pre-clinical material, specific content learning related to asthma and clinical knowledge, and collaboration between peers and faculty. The positive student evaluations spurred further development of the case and delivery to a larger cohort of the class.

VIP has gone through several iterations as it has been developed, both to pivot to an online format in response to the pandemic, further discussed below, and to adapt to student and faculty feedback, discussed in Findings.

**LEARNING OBJECTIVES**

**At the completion of this event the student will have/be able to**

**Anatomy**
1. Review the structures that make up the upper and lower airways and the structure that divides them
2. Describe the anatomical structures affected by asthma and contrast them with structures affected by other respiratory illnesses like rhinovirus or pneumonia

**Allergy/Immunology**
1. Explain the sequence and manifestations of Type I Hypersensitivity
2. Describe the immediate and late phases of allergy
3. Explain asthma as a manifestation of allergy

**Biochemistry**
1. Given a patient presenting with a severe asthma attack, identify the major factors that can contribute to mixed acidosis.
2. Given a patient that with a lactic acidosis secondary to a hyperadrenergic state, provide a biochemical model that integrates lipid and carbohydrate metabolism in liver, adipose and muscle tissue to account for the metabolic acidosis.

**Microbiology**
1. Describe epidemiology of respiratory viruses as a cause of asthma exacerbations
2. List respiratory viruses commonly causing asthma exacerbations
3. Analyze the pathophysiology of why certain respiratory viral infections increase the risk of asthma exacerbations and secondary bacterial infections
4. Describe classic presentations of respiratory viral infections in a child with asthma
5. Describe and compare commonly used diagnostic tests for respiratory viruses
6. List available therapeutic options for common respiratory viruses and their indications in children
7. Analyze what system-based practices can be taken in a pediatric practice to reduce the risk of respiratory viral infections and their complications in children with a known diagnosis of asthma

**Physiology**
1. Explain the mechanisms of action and potential side effects of medications used in asthma management
2. Interpret pulmonary function tests (PFTs) and compare normal PFTs with those observed in obstructive lung disease
3. Interpret arterial blood gas measurements in the context of a patient's course and make decisions about subsequent patient care

**Pharmacology**
1. Provide rationale for using bronchodilators and steroids during acute asthma
2. Describe the side effects of frequent administration of Albuterol

**Social and Family Determinants**
1. Recognize the psychosocial impact of asthma on pediatric patients and their families
2. Assess addressable asthma exacerbation risks and educational needs for patients and families
3. Recognize inequities, and our own assumptions/biases, in assessing and engaging with patients and families

**Figure 2.** The learning objectives for the vertical integration case are listed and categorized by academic domain.

- Prior to the live event, students are given the learning objectives and a PowerPoint presentation which includes the case presentation, including history, social history, review of systems, triage vitals, physical exam, and a step -by-step description of his emergency department and hospital course and plans for discharge and follow-up. Prompts and questions in the domains of the learning objectives that require students to apply their current experience in clinical practice in their pediatrics clerkship are inserted throughout the presentation to help prepare students for small group discussion during the live event. (See Supplementary File S1).
- The live event begins with an introduction to the case and a discussion of expectations for all student participation.
- Students are then divided into 6 randomly assigned breakout groups of 8 students corresponding with learning objectives (Microbiology-Immunology; Biochemistry; Physiology-Pharmacology; Social Determinants-Psychosocial issues; Clinical Management). Faculty rotate for 15 min of presentation and discussion facilitation in their area of expertise around the previously provided questions. During the small group sessions students are encouraged to discuss the case in line with the learning objectives (see Supplementary Table S1), reflect on what stood out for them in reviewing the case, what additional questions it raised, and what assessment or actions they or an interprofessional colleague might take in response.
- The event closes with a 20–30 min wrap-up where students return to a large group setting to share their experience, including having each breakout group responsible for reporting on one of the topics discussed.



In the next phase of the event development the case was delivered to groups of 48 students, which were roughly half of the Class of 2022. To accommodate the larger number of students and reduce redundancy, the event was modified by combining some of the content areas into single discussion groups. The disciplines of physiology and pharmacology were combined as were infectious disease and immunology. Anatomy, biochemistry, clinical management, and behavioral and social science content remained individual sections. The case was introduced to all forty-eight students who were then divided to six groups of eight students. Faculty rotated among the small groups so that every group met with all faculty members (see Figure 3). This process maintained the small group experience but had the disadvantages of requiring the faculty facilitators to review the identical material with each group over the course of the event (total of six times). It also led to each group discussing a topic in a different order, so that for one group clinical management preceded physiology while for another group pharmacology was discussed before biochemistry. The students' feedback mentioned this point; in order to maximize the educational benefits of working in small groups, we did not see another way of conducting the program.

| Flow of Events | Time Course | Group 1 | Group 2 | Group 3 | Group 4 | Group 5 | Group 6 |
|---|---|---|---|---|---|---|---|
| Pre-work: Case Review by Students Before Event | 60 min | X | X | X | X | X | X |
| Event Introduction and Instructions: Large Group | 15-20 min | X | X | X | X | X | X |
| Rotation 1 | 15 min | A | B | CM | MI | PP | SDPI |
| Rotation 2 | 15 min | SDPI | A | B | CM | MI | PP |
| Rotation 3 | 15 min | PP | SDPI | A | B | CM | MI |
| Break | 10 min | X | X | X | X | X | X |
| Rotation 4 | 15 min | MI | PP | SDPI | A | B | CM |
| Rotation 5 | 15 min | CM | MI | PP | SDPI | A | B |
| Rotation 6 | 15 min | B | CM | MI | PP | SDPI | A |
| Large Group Debrief: each breakout group reports for 5 min on 1 topic | 30 min | Report on MI | Report on CM | Report on A | Report on SDPI | Report on PP | Report on B |

**Figure 3.** Timetable for Event. X, large group of 48 students; A, Anatomy; B, Biochemistry; CM, Clinical Management; MI, Microbiology/Immunology; PP, Physiology/Pharmacology; SDPI, Social Determinants/Psychosocial issues.

In the most current iteration of the VIP event, to allow for more time for student engagement in the different disciplines, the following adjustments have been incorporated.

- A brief 5–10 min introduction is followed by a full-group review of asthma-related anatomy for an additional 20–25 min. Students review the anatomical structures that are part of the upper versus lower airways, describe which asthma primarily affects, and contrast wheezing versus stridor. They also review the boundary between conducting and respiratory airways, the gross appearance of the airways on a normal chest x-ray, and briefly describe the timing of the embryological and postnatal development of the lungs and alveoli.
- Students are then divided into 4 randomly assigned breakout groups of 12 students: Microbiology-Immunology; Biochemistry-Physiology-Pharmacology; Social Determinants-Psychosocial Issues; and Clinical Management. Faculty members rotate for 30 min of presentation and discussion facilitation in their area of expertise around the previously provided questions. Because of the combination of academic and clinical disciplines, the majority of rooms have 2 faculty facilitators. As in previous iterations, students evaluate the case in line with the learning objectives (Supplementary

Table S1), reflect on their thoughts and questions, and discuss what assessment or actions they or an interprofessional colleague might take in response.

## 3. Results

Even though each session is relatively short, students find it useful to have dedicated time with faculty with content expertise and experience, and to focus on "the most important points and big takeaways from the case rather than getting bogged down in minutia or tangents", as one student commented. The depth and variety of topics raised in these discussions are valued by students, as seen in the feedback on "how socioeconomics/other demographics affect health", "how SHD contribute to asthma risk", "exploring a multidisciplinary approach", "seeing the interplay between specialties over time", and integrating specialties "rather than keeping them siloed".

Evaluation of the larger student cohort was performed with the student evaluation form that was used with the pilot group with small modifications; thirty of the forty-eight students completed the evaluation (Figures 4 and 5). As with the pilot group, the larger cohort reported very positive experiences. Twenty-six students (87%) found the experience worthwhile, twenty-four students (80%) reported increased knowledge, and 73% said they would welcome additional similar events. While a preference for the small group setting declined when compared to the pilot group, the larger cohort (73% of responders) still preferred the small group setting. The narrative responses to the three valuable aspects of the event were very similar to the pilot group with the major themes being review of preclinical content, collaboration between peers or with facilitators, and learning of specific content. Suggestions for improvement had some similarities between the pilot and large cohort group, with both groups desiring additional resources to be used outside the event either in the form of a prereading or pre-assignments or a list of major learning issues to takeaway. Five (16%) students in the large cohort of students did suggest that the focus should be more clinical while fourteen (46%) listed the review of the basic science material as a valued aspect. In conclusion, the students embraced the vertical integration event and found it meaningful in a variety of aspects.

| Item | Pilot (8 of 8) | Full Cohort (30 of 48) |
|---|---|---|
| Worth the time | 8 (100%) | 26 (87%) |
| Increased Knowledge | 8 (100%) | 24 (80%) |
| Helped with Integrated Illness Script (1-5) | 4.8 | 3.7 |
| Faculty Involvement | 6 (75%) Just Right; 2 a bit too little | 28 (93%) Just Right; 2 a bit much |
| Small Group Size | 8 (100%) prefer size | 22 (73%) prefer size |
| Structure helped Learning (1-5) | 4.5 | 4.0 |
| Individual Input valued | 100% All the time or >50% | 87% All time or >50% |
| Rate the following: (1-5) Interacting with Facilitators | 4.3 | 4.1 |
| Collaborating with Peers | 4.4 | NA |
| Engaging with Material | 4.3 | 3.9 |
| Welcome Similar Events | 100% Absolutely or In Select Topics | 73% Absolutely or In Select Topics |
| Suggestions for Improvement | Handout summary Release Topic Reduce Redundancy | More Clinical and Less Preclinical Content Timing/Ordering of Content Pre-Rereading or Major Takeaways Small Groups versus Large Groups |
| Three Valuable Things | Integration of Pre-clinical and Clinical Review of Pre-clinical Content Specific Knowledge (Asthma or Clinical Knowledge) Collaboration | Review of Pre-clinical Collaboration Small Group and Content Specific Knowledge |

**Figure 4.** Summary of results from student evaluations of pilot and full cohort VIP.

While student satisfaction is an important component, effective learning events must advance the learner's career preparation. The Accreditation Council for Graduate Medical Education (ACGME) is the body responsible for accrediting all graduate medical training programs for physicians in the United States. The six ACGME Core Competencies are as follows: Practice-Based Learning and Improvement, Patient Care and Procedural Skills,

Systems-Based Practice, Medical Knowledge, Interpersonal and Communication Skills, and Professionalism. These competencies are expectations for medical students graduating from medical schools in the United States [16].

| Item | Selected Comments |
|------|-------------------|
| Worth the time | I enjoyed this session more than traditional didactic sessions in which it is just a lecture with no audience participation. |
| | Yes, but the order of the sessions is important. As an example, our group had the anatomy session last which we think would have been most valuable in the beginning to support the other lectures |
| Strengths and Weaknesses | Wonderful integration to clinical work! |
| | 10/10 Loved this session. |
| | This was great! As third years, it is important for us to have a clinician in each session. |
| | I found this to be a great session, little by little building/recalling and combining our knowledge. |
| | The pharmacology was helpful.... |
| | Love them! Good balance between teaching and interaction. |
| | Even though biochemistry is a nightmare Dr. D... and Dr. H... made it a dream. |
| | I appreciated the discussion questions in the PPT that accompanied the discussion. It helped direct the discussion about important considerations. |
| | Both were great at getting us to think about various steps of the immune response, what we would consider as pathogens, how we would treat. |
| | Really appreciated this session to consider other aspects of care we don't always cover. |
| Suggestions for Improvement | I appreciated the faculty engagement and felt that they were very invested in my education. I generally prefer workshop sessions with high levels of participation to be more beneficial than traditional didactic lectures. |
| Three Valuable Things | Biochemical pathways have utility even in clinical context breaking down a case into themes we wouldn't otherwise have thought about. |
| | Great review of material from 1st two years. |
| | Enhanced my knowledge of asthma |
| | enjoyed working with our previous staff again... |
| | Each individual session was valuable. |

**Figure 5.** Selected narrative from VIP student evaluations.

The Vertical Integration in Pediatrics session addressed the ACGME competencies as follows:

- Practice-Based Learning and Improvement: Students were required to evaluate patient care practices and appraise and assimilate scientific evidence based upon the case data to inform the course of care.
- Patient Care and Procedural Skills: Through their work with clinical faculty and a licensed social worker, students learned how providers should interact directly with the patients under their care.
- Systems-Based Practice: During the event, students were required to call effectively on other resources in the health care system, for example, social work, to provide optimal health care.
- Medical Knowledge: Students were required to demonstrate knowledge and retrieve prior knowledge in the disciplines of anatomy, biochemistry, pharmacology, physiology, microbiology, and immunology.
- Interpersonal and Communication Skills: Students worked with their peers to address posed questions and to collaborate on how to provide appropriate care. Students were coached on the best practice for communication with the case patient and family.
- Professionalism: Students met the expectation of medical professionals to treat people with respect, compassion, and dignity by coming prepared to the session, working cooperatively with peers and faculty, and by learning the best practices for professionalism in patient and family interactions from clinical and social sciences faculty.

## 4. Discussion

The Flexner report published in 1910 helped shape medical education in the United States [1,2] and remained the "gold standard" for American medical education throughout the twentieth century. Towards the end of the twentieth century medical schools have moved towards an integrated curriculum, shifting from the traditional 2 + 2 curriculum to a Z-shaped curriculum [8]. In the Z-shaped curriculum, clinical cases are incorporated with basic sciences starting at the early years, with increased attention to clinical skills. In this curriculum, basic sciences teaching gradually decreases while the intensity of clinical teaching ramps up. The rationale for an integrated curriculum is based in part on instructional design theory, which advocates training students in an integrated curriculum

with whole tasks rather than separate ones [17]. Integrating the curriculum has other benefits including fostering students' social identification with the community of practicing physicians, thus contributing to professional identity formation [18].

Other benefits of integrating clinical education earlier in medical school include support for the students' ability to work independently, solve medical problems, manage unfamiliar situations, prioritize tasks, and collaborate [6]. Similar benefits were also found by Littlewood et al. [19] who conducted a systematic review of the literature dealing with the implementation of early clinical experience. The studies they analyzed found that students who experienced early clinical experience were more satisfied with their studies. The authors also found data suggesting that early clinical experience improved students' ability to relate to patients, communicate empathy, take better history, and perform simple physical examination. Overall, the students felt better able to approach patients.

Based on the broad literature suggesting the many benefits of an integrated curriculum, the educational leaders of the Netter School of Medicine looked to modify the traditional 2 + 2 curriculum into an integrated one. Such curricular changes require time to implement with the involvement of many stakeholders. While discussions were on-going regarding changes in the curriculum, a group of faculty members met with the intention to develop a pediatric clerkship-specific integrated educational program. It was clear that the program's success required the full cooperation of the faculty, who were already working extended hours, and teaching students from years one and two. Delivering the session in-person rather than online also contributed to the time constraints. Due to the pandemic, the sessions were later conducted via zoom and were repeated a few times in the academic year, requiring the faculty to carve out time from their regular schedules and dedicate it to the program.

The majority of the students enjoyed meeting again the year one and two faculty. The discussion of basic science concepts in proximity to reviewing clinical management enabled the students to better understand the various facets of the case, including pulmonary physiology, biochemistry of blood gases, effects of medications on lung function and on blood chemistry, which viruses and bacteria initiate the cascade of respiratory deterioration in asthma, and the immunological aspects of the disease.

Recognizing and mitigating the social determinants of health (SDH) and supporting health equity are now recognized as vital components of graduate medical education [20]. Medical students may feel that a deeper consideration of SDH is key to their learning and may be insufficiently covered in didactic curricula. Our case study allows students to connect "micro" and "macro"—the biopsychosocial interplay centering in this moment on the patient and his asthma [21]—within the context of structural, social, political, and environmental determinants [22]. This dual approach provides important clinical lenses to the VIP, moving students beyond the basic science and clinical management of the case to thinking about a direct approach to patient- and family-centered care and to the larger systems that patients and families live within. It was especially encouraging to uncover the students' perceived benefits from the discussion of the effects of a chronic illness in a child on the rest of the family. Such benefits validated that including social work faculty in the VIP led the students to have a better understanding of the psychosocial aspects of pediatric disease and the role of the family in the child's health.

The students provided very few negative comments. One example was the request to receive information about the case and learning objectives prior to the session so that they could arrive prepared for the small group discussions; this modification was made to the most recent iteration, and faculty felt that the students were more prepared for the discussions. It was encouraging to find out that many students said they would like to see similar programs in other clerkships. While the expansion of similar events to other clerkships is desirable, the availability of faculty to attend the session in-person places more pressure on already tight schedules. The hope is to move towards integrated curriculum across the school, by adding similar programs that can be organized and synchronized between clerkships and years one and two faculty.

Team based learning (TBL), case based learning (CBL), and problem based learning (PBL) are common approaches used in medical education to achieve horizontal and vertical integration among the pre-clinical curriculum. These methods require learners to retrieve aspects of the year 1 and year 2 didactic curriculum, develop skills in collaborative and self-directed learning, and apply the knowledge for use in clinical contexts [23]. As TBL, CBL, and PBL are typically conducted in the pre-clinical years, students are challenged to extrapolate to clinical treatments because they lack the context of full-time medical practice. Our curriculum builds on these established methods by requiring 3rd year medical students in their clinical rotations to integrate what they are learning in hands-on patient care with the basic and social science aspects of biology of disease, diagnosis, and treatment. A recent study using an analogous model in a 3rd year surgery rotation found that knowledge scores were high after the event and when assessed several weeks afterwards, suggesting effective and long-term knowledge transfer [24].

While successful in implementation, there are limitations to the current study. A pre-event and post-event student assessment could have provided data to evaluate the growth in student skills by integrating basic science and clinical application. Student engagement may have also been impacted by the virtual format, as the interactions between faculty and students that normally occur in person were lacking. Finally, it would have been valuable to have a tool to measure whether the event improved student performance in their clinical patient encounters.

In summary, the feedback from the students and the readiness of our faculty to invest time and energy in the program suggested that the program has merit and will therefore continue.

## 5. Conclusions

Medical schools are looking into moving from the traditional 2 + 2 curriculum to an integrated one. Clinical experience is introduced during the period of teaching basic sciences, followed by a simultaneous decrease in teaching basic science with an increase in clinical responsibilities. Data suggest the reinforcement of basic science material in the context of clinical application (Z-shaped curriculum) benefits the students both short and long term. At the Netter School of Medicine, a pediatric clerkship vertical integration program was developed, which is taught concurrently by basic science and clinical faculty with collaboration with a professor from the department of social work. In this Z-shaped integrated program, students review basic science material, while at the same time learning about clinical management and becoming aware of the effects of chronic illness on the family. The Z-shaped curriculum model was emphatically accepted by students, who expressed interest in having such programming in other clerkships. The Netter experience suggests that in-person clinical review during clerkships in year three, combined with basic science in-person teaching by the same faculty from years one and two has merit and should be considered in other programs.

**Supplementary Materials:** The following supporting information can be downloaded at: https://www.mdpi.com/article/10.3390/educsci13060545/s1, File S1: Case Synopsis and Guiding Questions; Table S1: Mapping of Learning Objectives to Embedded Student Questions.

**Author Contributions:** All authors were involved in conceptualization and writing of the manuscript. J.M.H. was lead on writing and editing. E.S.K. was the project director. All authors have read and agreed to the published version of the manuscript.

**Funding:** This research received no external funding.

**Institutional Review Board Statement:** Ethical review and approval were waived for this study due to focus on quality improvement project related to curriculum change and student feedback.

**Informed Consent Statement:** Not applicable.

**Data Availability Statement:** Not applicable.

**Acknowledgments:** We would like to express our sincere gratitude to Pediatric Clerkship Coordinator, Casey LoRusso.

**Conflicts of Interest:** The authors declare no conflict of interest.

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
