# Peer review of "Vertical Integration in the Pediatrics Clerkship: A Case Study"

_education, doi:10.3390/educsci13060545_

Round 1

Reviewer 1 Report (Previous Reviewer 2)

Basically, I think of this as a valid and promising manuscript, however, changes done thus far haven’t actually made it up to scratch –

-        I suggest these papers way too often (maybe), but I assure you that they are useful and very crucial for the field of vertical integration -  Wijnen-Meijeret al. [CrossRef] [PubMed] and   Rosenthal et al.. [CrossRef]

-        “figure 2” should be a “table”; also, learning objectives should be listed with more care; list them attentively and explain logically; maybe you could reorganize the whole clerkship (oh, what an unfortunate term!) so that you have few basic courses – classroom education and guidance, common to other medical professions too (basis of a “zed”) – not just fundaments for pediatric clerkship; in that sense, you should correct the diagram’s text (referring to” figure 1” here).  I like the phrase “basic & classroom teaching” more – when you say “basic science”; I expect kids to experiment. 

Author Response

We thank the Reviewer for their time, careful and thoughtful review, and highly constructive comments. We have addressed all of the Reviewers comments below.

Reviewer Query: I suggest these papers way too often (maybe), but I assure you that they are useful and very crucial for the field of vertical integration -  Wijnen-Meijeret al. [CrossRef] [PubMed] and   Rosenthal et al. [CrossRef]

 Author Response: We thank the reviewer for suggesting useful references in the field of vertical integration. In the original manuscript, we had included 2 references from Wijnen-Meijeret et al. Unfortunately, the hyperlink for the recommended Rosenthal reference did not transfer in the review, and we had difficulty uncovering which reference was meant. With Rosenthal being a common name, we uncovered a few references in the field of Vertical Integration, but we were not sure which one was being referred to.

Reviewer Query: Figure 2 should be a table.

Author Response: We have addressed the Reviewer’s suggestion by renaming Figure 2 as “Table 1,” and renumbering other tables in the legends and text.

Reviewer Query: Learning objectives should be listed with more care; list them attentively and explain logically.

Author Response: The Learning Objectives are currently listed in categories of academic domain so that students have a framework for the expectations for learning in various disciplines of basic science classroom instruction. There are many Learning Objectives, because we wanted to be very specific about what our expectations for learning. We would benefit from clarification of the Reviewer’s comment in order to make any changes.

Reviewer Query: Maybe you could reorganize the whole clerkship (oh, what an unfortunate term!) so that you have few basic courses – classroom education and guidance, common to other medical professions too (basis of a “zed”) – not just fundaments for pediatric clerkship.

 Author Response: We absolutely agree that Clinical Clerkships would greatly benefit from the inclusion of a few basic classroom courses. All Clerkships do currently have some didactic components, but regrettably, very little as far as basic teaching and learning. In order to meet the Learning Objectives of the US Medical School accrediting body (the Liaison Committee on Medical Education for American Colleges) and the residency standards from the Accreditation Council for Graduate Medical Education, the majority of the time in the Clerkships must be devoted to achieving clinical competencies. Our vertical integration curriculum is one step in the movement to add more basic and classroom teaching in the context of clinical care.

Reviewer Query: Maybe you should correct the diagram’s text (referring to” figure 1” here).  I like the phrase “basic & classroom teaching” more – when you say “basic science”; I expect kids to experiment. 

 Author Response: We have corrected the text in the Figure 1 legend to read “basic and classroom teaching.”

Reviewer 2 Report (Previous Reviewer 1)

All my comments were answered.

Author Response

We thank Reviewer 2 for their time, careful review, and especially for lending their expertise to evaluating our manuscript. Reviewer 2 states that all of their comments have been answered/addressed. Thank you again for your thoughtful review and efforts toward greatly improving the manuscript.

This manuscript is a resubmission of an earlier submission. The following is a list of the peer review reports and author responses from that submission.

Round 1

Reviewer 1 Report

The title does not sustain completely the content of the manuscript. It should also reflect the educational feature.

Introduction. Include also the part of medical education in pediatrics field.

Discussion. The paragraph related to the study results (lines 275-287) repeats. It can be removed, except for the last part.

What are the limitations of the present study?

Conclusions. The conclusions should be rephrased in order to point out the main findings. The Z-shaped integrated curriculum should be strengthened.

Reviewer 2 Report

  1. What was  the  definition  of  integrated case review?    Were eycases also included in this study?
  2. What was the definition of  competency in medical education based on vertical integration didactic session?
  3. Disclose conflicts of interest of the  authors and   mention informed  consent  from  participants/ IRB  Were the participants given  the  option  to  undergo  conventional approach?  

Reviewer 3 Report

This manuscript describes a teaching/learning activity, which integrates basis medical sciences in clinical case description. Although the description is interesting, improvements are necessary to make this manuscript acceptable for publication. Because the manuscript describes a relatively short educational intervention (2 hours?) the authors could better concentrate on a reflective evaluation of their case, rather than spend a lot of text on repeating alredy published discussions about “integration”.

Major remarks:

  1. The authors go a long way in describing the potential advantages of (what they call) vertical integration, both in the Introduction and the Discussion section, but fail to interpret the results of their own case study in terms of these educational aspects of integrated teaching/learning activities (TLA). Their case study is only evaluated on the lowest level of Kirkpatrick’s evaluation pyramid (i.e. student satisfaction). This reviewer, therefore, suggests that the description of the potential advantages of integrated TLAs is reduced in size and that more attention is paid to existing education literature, which describes concrete examples of similar TLAs in medical education. In the discussion their case study could be evaluated in relation to other, similar but not neessarily identical, designs of content integration.
  2. In order to understand the actual design of the TLA more details are required, both with respect to timing of the “live event”, and with respect to the way content from different disciplines is presented or explored by the students. From the description under Methods it appears that the TLA starts with a full group session (How long? What is the role of teachers? What is the role of teachers? Which teacher?), followed by break-out sessions (30 minutes in total? 6 groups of 8 students?). From Supplement A it appears that the information for students (starting from the sentence “The case follows the course of Marcus, an 8-year old male child ...”) is very well structured and reads like a sequential set of assignments which can be solved by any advanced student. What is the added advantage of the break-out sessions? Finally, it is not clear which preparation (i.e. before the live event) is required of the students? For the reader it would be more informative to have a short description of both the preparatory activities and the timing of the live event in a Table format, including time frame and a description of the teachers’ disciplinary content at various stages.
  3. The authors are to be commended for the decription of the learning goals of the TLA (figure 2), but it would be helpful make visible how these learning goals are embedded or used in the TLA. This could be done in the form of a matrix of learning goals versus student-oriented questions (in the Appendix), or similar design elements of the TLA.

This reviewer regrets to be this critical. The manuscript describes an interesting idea for development of integrated TLAs in medical education, but a more detailed description and evaluation may be necessary to illustrate the full potential of the design of this TLA. A more detailed description and evaluation could also be helpful for generating other cases, beyond pediatric airway-problems.

Reviewer 4 Report

Overall, this paper was well written and interesting to read. However, the introduction was too long and repetitive and the whole paper would be much better if approached as a short report or case study. There is insufficient ‘new knowledge’ to merit a full paper and a superficial review of the literature considering how long the introduction was.

The initiative and methodology are sound, and it is a great initiative, but nothing about it is new in the field of Medical Education. Integrated curriculum is the expected norm in medical education now.

Team based learning (TBL), Case Based Learning (CBL), and Problem based learning (PBL) are all elements which help to vertically and horizontally integrate curricula and yet these weren’t mentioned anywhere in the Introduction. The 7 paragraphs taken to explain VI and integrated curricula could have been done in 1-2 paragraphs, there is no doubt about integrated curricula.

There was no clear rationale for the study or research question stated for the investigation.

Materials and Methods

  • The Learning Objectives in Figure 2 are excessive for a single learning session. If the session was evidence of integration than surely one element of this would be a reduced number of learning objectives?
  • It sounds like the integration occurs via a single event – I am wondering where the consolidation of learning happens? Upon further review of the whole method it actually sounds like an exemplar of Case Based Learning
  • It is unclear as to whether each of the rotations in the pilot group comprised just 2 students. If key faculty and clinicians rotated through the 48 groups in the follow up event, and that was based on the pilot, was the same process followed?

Although the need to revisit basic sciences material in the clinical years is recognized, and the strategy described in the paper is commendable, it would have been useful to see what the flow on effects were. For example, did it improve student interactions with patients? Did it enhance their clinical reasoning? Did their clinical supervisors notice a difference in performance? As it stands, this is a straight evaluation of a nicely implemented learning activity but not really anything that is innovative, adds new knowledge to the field, and demonstrates impact on learning and performance.

The materials and methods may be worthwhile submitting to the MedEdPortal for others to consider adopting at their own sites. https://www.mededportal.org/